# OpenReview forum: "GePBench: Evaluating Fundamental Geometric Perception for Multimodal Large Language Models"
_ICML.cc/2026/Conference — ICML 2026 regular_

### Official Review · Reviewer_nLRJ · 2026-03-01

**Soundness:** 3
**Presentation:** 3
**Significance:** 3
**Originality:** 3
**Overall Recommendation:** 4
**Confidence:** 3

**Summary:**

The paper introduces GePBench, a large-scale, synthetically generated benchmark containing 80K images and 285K multiple-choice questions across six dimensions (existence, counting, location, size, reference, and relationship). The paper evaluates 27 MLLMs, revealing significant deficiencies, particularly in size and location estimation. Through in-depth diagnostic experiments (linear probing, saliency maps, and module-specific fine-tuning), the authors discover that the LLM backbone, rather than the vision encoder, is the primary bottleneck causing spatial information loss. Finally, they train a model (LLaVA-GeP) using their synthetic data engine, demonstrating that enhancing basic geometric perception transfers to consistent improvements on downstream real-world tasks like medical VQA, math, and chart understanding.

**Compliance With Llm Reviewing Policy:**

Affirmed.

**Key Questions For Authors:**

No

**Limitations:**

No discuss limitations, can refer to weaknesses for suggestions.

**Strengths And Weaknesses:**

**Strengths**
1. The motivation is fundamental and paper highly support the motivation.
2. The paper is well-structured, comprehensive in its evaluation (27 models + human baselines), and easy to follow.
3. The diagnose experiment finds out the root of information loss is the LLM backbone instead of visual backbone is interersting.

** Weaknesses**
1. Though the paper targets that the cause of geometric information, the paper needs an analysis why this happens, for example, it does not explore whether this loss of spatial awareness is due to the 1D serialization (flattening) of 2D image tokens,.

---

> ### Author Rebuttal · Authors · 2026-03-31
>
> Thank you for your appreciation of our fundamental motivation, comprehensive evaluation and insteresting disgnostic analyses. For the concerns, we detail our response below. We would appreciate it if you could let us know whether your concerns are addressed.
>
> > ***Q: Though the paper targets that the cause of geometric information, the paper needs an analysis why this happens, for example, it does not explore whether this loss of spatial awareness is due to the 1D serialization (flattening) of 2D image tokens.***
>
> **A:** This is an insightful question. In Appendix D, we present a detailed error analysis tracking the source of model failures, concluding that the core bottleneck lies within the LLM backbone rather than the visual encoder's 1D serialization.
>
> Modern MLLMs utilize 2D Rotary Positional Encoding (RoPE), which successfully preserves and injects 2D structural relationships into the sequence. Therefore, the architectural flattening of tokens is not the primary culprit. Instead, our findings align with recent studies [1, 2] indicating that highly structured visual tokens are gradually overridden by the strong language priors inherent to the LLM backbone. When confronted with purely visual, geometry-heavy inputs lacking dense semantic text (as in GePBench), the LLM struggles to route attention effectively, leading to hallucinations or random guessing.
>
> While fully resolving this visual dilution is an open problem in multimodal architecture (potentially requiring end-to-end native modeling [3] or novel attention mechanisms [4]), GePBench serves as the exact foundational testbed needed to quantify and ultimately fix this phenomenon. We will incorporate this specific architectural discussion into the main text to strengthen the paper's analytical depth.
>
> [1] Is A Picture Worth A Thousand Words? Delving Into Spatial Reasoning for Vision Language Models
>
> [2] Decoupled Proxy Alignment: Mitigating Language Prior Conflict for Multimodal Alignment in MLLMs
>
> [3] Gemini 2.5: Pushing the Frontier with Advanced Reasoning, Multimodality, Long Context, and Next Generation Agentic Capabilities
>
> [4] Perceiving Beyond Language Priors: Enhancing Visual Comprehension and Attention in Multimodal Models

---

> > ### Author Rebuttal · Reviewer_nLRJ · 2026-04-04
> >
> > Thanks for the rebuttal, I decide to keep my positive score.

---

### Official Review · Reviewer_Bq7o · 2026-03-11

**Soundness:** 3
**Presentation:** 3
**Significance:** 3
**Originality:** 3
**Overall Recommendation:** 4
**Confidence:** 4

**Summary:**

This paper introduces GePBench, a new benchmark for evaluating MLLM fundamental geometric perception. The benchmark concentrates on lower-level abilities such as existence, counting, size, location, reference, and relationship understanding over geometric figures, instead of high-level geometric tasks. Experiments show that even strong models still perform poorly on several basic geometric perception tasks, with especially weak results on size and location. Fine-tuning LLaVa-GeP shows that improving geometric perception can improve downstream multimodal task performance.

**Compliance With Llm Reviewing Policy:**

Affirmed.

**Final Justification:**

The authors addressed all my concerns in the rebuttal.

**Key Questions For Authors:**

1. Can the authors clarify whether the multiple-choice format may overestimate model capability (compared with other forms such as open-ended QA)? It is unclear whether the performance reflects true geometric perception or option-matching behavior.

2. In LLaVA-GeP, both caption-style pretraining data and multiple-choice instruction data are added. Which part contributes most to the downstream performance improvements?

3. Can the authors clarify the inconsistency in the human evaluation setup (20 participants in the main text vs 25 volunteers in the appendix)?

**Limitations:**

The synthetic and multiple-choice nature of the benchmark, and the limited originality, as discussed in Strengths and Weaknesses.

**Strengths And Weaknesses:**

Soundness.
The paper is technically solid overall. The benchmark construction pipeline is clearly motivated. Experiments are abundant to support the opinions.

The main limitation is that the benchmark is highly synthetic and based on a multiple-choice form, which cannot fully capture open-ended geometric understanding. Also, the fine-tuning part uses LLaVA family as base model, which is relatively outdated and performs worse than recent MLLMs such as Qwen3-vl or Qwen2.5-vl.

Presentation.
The paper is generally well organized and easy to follow. The difference between basic geometric perception and geometric reasoning is clearly defined, leading to a more straightforward understanding. The figures and tables are helpful. One issue is that the main paper mentions 20 participants, while the appendix mentions 25 volunteers, which should be corrected.

Significance.
The problem is important. Many multimodal tasks rely on fine-grained spatial and structural understanding, yet most current evaluations focus on higher-level semantic reasoning, ignoring basic geometric tasks. The downstream fine-tuning results are also valuable as improving basic geometric perception leads to performance gain on downstream task perception.

Originality.
While the paper seems to be the first to evaluate MLLMs on basic geometric tasks, integrating surrogate geometric tasks or fine-tuning with geometric tasks is not a brand new perspective. Some previous papers also tried to improve spatial understanding via fine-tuning with geometric tasks[1][2].

[1] Liu, Daixian, et al. "TangramPuzzle: Evaluating Multimodal Large Language Models with Compositional Spatial Reasoning." arXiv preprint arXiv:2601.16520 (2026).
[2] Lian, Shijie, et al. "Euclid's gift: Enhancing spatial perception and reasoning in vision-language models via geometric surrogate tasks." arXiv preprint arXiv:2509.24473 (2025).

---

> ### Author Rebuttal · Authors · 2026-03-31
>
> Thank you for your appreciation of our technical solidness, well-organized presentation, and important research question. For the concerns, we detail our response below. We would appreciate it if you could let us know whether your concerns are addressed.
>
> $~$
>
> ---
>
> $~$
>
> > ***(Q1 & W1) Question form: multiple choice v.s. open-ended***
>
> **A:** We utilized a multiple-choice format because it provides an objective, standardized evaluation metric, a standard widely adopted in both human assessments (SAT, GRE) and leading multimodal benchmarks (MMMU-Pro, MMBench).
>
> To rigorously test whether the multiple-choice format artificially inflates performance, we converted a subset of GePBench (2k samples) into open-ended, generative questions. For Reference and Existence, options are part of the question itself and cannot be removed, so this analysis only includes the following 4 aspects:
>
> |Aspect|Open-ended question form|Evaluation creteria|
> |-|-|-|
> |Counting|Ask for the exact number directly|Exact integer match|
> |Size|Ask for the numeric value in decimal|Value within a strict error margin|
> |Location|Ask for the centroid coordinates|Coordinates fall within the correct quadrant|
> |Relationship|Ask to describe the relationship|Semantic match of correct relationship|
>
> As shown in the new results table, all models exhibit a moderate, expected performance decline across all aspects when transitioning to open-ended questions. This drop reflects the inherent difficulty of generating precise content rather than a failure of the benchmark format itself.
>
> |Model|Multiple Choice|Open-ended|Difference|
> |-|-|-|-|
> |gpt-4o|59.0|49.5|-9.4|
> |Qwen2.5-VL-7B|59.1|39.5|-19.5|
> |Qwen2.5-VL-72B|63.0|54.9|-8.1|
> |InternVL3-8B|50.0|36.8|-13.3|
> |InternVL3-78B|66.3|48.7|-17.6|
>
> $~$
>
> ---
>
> $~$
>
> > ***(W1) Newer base models for LLaVA-GeP***
>
> **A:** We extended our LLaVA-GeP methodology to the modern Qwen2.5-VL and Qwen3-VL architecture. Since their training data are not publicly available, we use the LLaVA-1.5 training data (mixed with our synthethic geometric data) to train the models from scratch. As summarized in the table, the geometric perception enhanced models exhibit considerable gains across diverse downstream tasks. This proves that our proposed data synthesis engine is model-agnostic and that geometric perception remains a universal bottleneck across different generations of architectures.
>
> |Model|Average|General VQA|Downstream Application|
> |-|-|-|-|
> |LLaVA-1.5-7B|49.2|61.5|36.8|
> |LLaVA-GeP|50.7|62.7|38.8|
> |Qwen2.5-VL-LLaVA-7B|56.9|69.1|44.7|
> |Qwen2.5-VL-GeP|57.7|69.5|45.9|
> |Qwen3-VL-LLaVA-8B|63.5|72.9|54.1|
> |Qwen3-VL-GeP|64.8|74.2|55.4|
>
> $~$
>
> ---
>
> $~$
>
> > ***(Q3 & W2) Participant number mismatch***
>
> **A:** We sincerely apologize for this typographical error. The correct number of participants is 25. We expanded the human evaluation pool to ensure broader demographic coverage but missed updating the integer in the main text. This will be corrected in the revised manuscript.
>
> $~$
>
> ---
>
> $~$
>
> > ***(W4) Previous papers on spatial understanding***
>
> **A:** We agree that the two papars are excellent works targeting general spatial awareness, and we will properly contextualize them in our Related Work. However, spatial awareness is only one facet of geometric perception. While these existing works largely focus on spatial folding or abstract coordinate tracking, GePBench targets the geometric shape itself. Beyond location and size, our benchmark enforces rigorous evaluation of existence, counting, reference, and structural relationships. This ensures a holistic assessment of an MLLM's ability to identify objects, recognize shapes, and understand physical interactions, acting as a comprehensive proxy for real-world geometric understanding.
>
> $~$
>
> ---
>
> $~$
>
> > ***(Q2) Ablation on LLaVA-GeP***
>
> **A:** To isolate the contributions of different data, we conducted an ablation study on LLaVA-GeP. We trained models using only caption-style pretraining data and only multiple-choice instruction data respectively.
>
> As shown in our results table, while both configurations independently yield moderate downstream improvements, the combined dataset achieves the highest accuracy. The caption-style data establishes foundational visual-to-text alignment for shape features, while the instruction data forces the model to engage in comparative perceptual understanding. Both phases are crucial.
>
> |Model|Average|General Visual Capability|Downstream Application|
> |-|-|-|-|
> |LLaVA-1.5-7B|49.2|61.5|36.8|
> |LLaVA-GeP (caption)|49.4|61.3|37.5|
> |LLaVA-GeP (QA)|50.2|62.4|37.9|
> |LLaVA-GeP|50.7|62.7|38.8|

---

> > ### Author Rebuttal · Reviewer_Bq7o · 2026-04-04
> >
> > I appreciate the authors' efforts, and all my concerns have been adequately addressed.

---

### Official Review · Reviewer_Lzvq · 2026-03-12

**Soundness:** 3
**Presentation:** 3
**Significance:** 2
**Originality:** 2
**Overall Recommendation:** 4
**Confidence:** 3

**Summary:**

The paper introduces GePBench, a large-scale synthetic benchmark specifically designed to evaluate the geometric perception of Multimodal Large Language Models (MLLMs).
The authors developed a specialized data synthesis engine that generates structured textual descriptions, renders them into figures, and produces 285K multiple-choice questions for 80K images. Evaluations of 27 state-of-the-art models (including GPT-4o, Gemini-2.5-Pro, and Qwen2.5-VL) reveal that MLLMs struggle significantly with these basic tasks—often performing below random guessing in size and location categories. Finally, the authors show that fine-tuning a model (LLaVA-GeP) on this data yields consistent performance gains in downstream domains.

**Compliance With Llm Reviewing Policy:**

Affirmed.

**Final Justification:**

The authors have provided detailed comparisons with previous related benchmarks, and conducted new experiments to extend their method to more recent models, like Qwen3-VL. The authors have successfully addressed my concerns during the rebuttal period. I will increase the score accordingly.

**Key Questions For Authors:**

Please refer to the weakness part.

**Limitations:**

yes

**Strengths And Weaknesses:**

Strengths:
- Soundness & Scale: The dataset is massive (285K questions) and includes a human-baseline validation (99.3% accuracy), proving that the tasks are trivial for humans but difficult for machines.

- Empirical Impact: The transfer learning experiments with LLaVA-GeP provide evidence that geometric perception is a foundational skill that enhances performance in real-world applications (medical, scientific documents).

Weaknesses:
- Significance: The findings that current state-of-the-art MLLMs exhibit significant deficiencies in geometric perception tasks are not novel. Similar conclusions are also shown in previous studies [1-2].

- Originality: As many previous studies also target geometric tasks, such as Geometry3K, it is necessary that the authors should list the detailed comparisons with them, instead of vaguely describing the comparisons in the Introduction part. This can help further highlight the uniqueness of the proposed benchmark.

- Experiments: The evaluated models in this paper are kind of outdated, such as Qwen2.5-VL, and the proposed LLaVA-GeP is also based on LLaVA-1.5-7B. To ensure that the conclusions still hold in SOTA MLLMs, it is necessary that the authors should employ more recent MLLMs for experiments.

[1] Euclid: Supercharging multimodal llms with synthetic high-fidelity visual descriptions. arXiv preprint arXiv:2412.08737 (2024).

[2] VisOnlyQA: Large Vision Language Models Still Struggle with Visual Perception of Geometric Information. In COLM 2025.

---

> ### Author Rebuttal · Authors · 2026-03-31
>
> Thank you for your appreciation of our soundness benchmark and empirical impact. For the concerns, we detail our response below. We would appreciate it if you could let us know whether your concerns are addressed.
>
> $~$
>
> ---
>
> $~$
>
> > ***Q1: Significance & Originality: The findings that current state-of-the-art MLLMs exhibit significant deficiencies in geometric perception tasks are not novel. Similar conclusions are also shown in previous studies [1-2]. As many previous studies also target geometric tasks, such as Geometry3K, it is necessary that the authors should list the detailed comparisons with them, instead of vaguely describing the comparisons in the Introduction part. This can help further highlight the uniqueness of the proposed benchmark.***
>
> **A1:** While existing works target geometry, they typically align more closely with mathematical problem-solving than with the spatial arrangements of shapes and their relationships emphasized by GePBench. In contrast, GePBench ensures targeted evaluation and offers comprehensive coverage of geometric shapes and relationships. To be specific, we summarize the key comparative aspects:
>
> |Dataset|Figure type|Task type|#shapes|#relationships|#samples|
> |-|-|-|-|-|-|
> |Geometry3k|Geometric diagram with notations|Mathematical problem solving|9|-|3K|
> |Euclid|Geometric diagram with notations|Size, relationship, reference|4|3|12K|
> |VisOnlyQA|Geometric diagram with notations|Size, existence|10|-|72K|
> |GePBench|Combination of pure geometric shapes|Counting, size, relationship, reference, existence, location|15|11|285K|
>
> 1. Task Scope: GePBench encompasses additional aspects of counting and location, enabling the assessment of accurate object identification and spatial awareness.
>
> 2. Figure Type: Prior benchmarks typically employ mathematical diagrams with geometric notations, while GePBench uses figures composed of pure geometric shapes. This design reflects real-world scenarios where annotations are often absent.
>
> 3. Diversity: GePBench includes more shapes and relationships, such as sectors and tangents, which frequently appear in practical domains like biological morphology and mechanical design.
>
> In summary, GePBench offers a targeted, comprehensive and realistic evaluation framework for geometric perception in MLLMs. Thank you for your suggestion and we will add the above comparison in detail in the related work section explicitely.
>
> $~$
>
> ---
>
> $~$
>
> > ***Q2: Experiments: The evaluated models in this paper are kind of outdated, such as Qwen2.5-VL, and the proposed LLaVA-GeP is also based on LLaVA-1.5-7B. To ensure that the conclusions still hold in SOTA MLLMs, it is necessary that the authors should employ more recent MLLMs for experiments.***
>
> **A2:** Thank you for the suggestion. To confirm the universality of our findings, we expanded our evaluation to include the most recent SOTA models, including GPT-5.2, Qwen3-VL, and Qwen3.5. As shown in the table, while newer and larger proprietary models show incremental improvements, smaller open-weight models fail dramatically at foundational geometric perception. Additionally, on specific aspects such as size, even the strongest models are still unable to achieve satisfactory results. These results emphasize that current MLLMs still lag behind humans by large margins.
>
> |Model|Size|Avg.|easy||||||hard||||||
> |-|-|-|-|-|-|-|-|-|-|-|-|-|-|-|
> ||||Ext.|Cnt.|Siz.|Loc.|Ref.|Rel.|Ext.|Cnt.|Siz.|Loc.|Ref.|Rel.|
> |Qwen3-VL|8B|67.8|77.4|83.1|22.4|63.8|86.9|89.5|69.2|66.8|29.4|73.7|80.2|70.6|
> |Qwen3-VL|235B-A22B|78.8|84.1|85.5|54.9|83.6|87.6|90.7|79.5|73.1|64.4|83.0|79.7|79.7|
> |Qwen3.5|9B|71.2|70.8|82.1|23.8|81.2|90.5|85.2|70.8|73.1|32.8|80.4|83.5|80.4|
> |Qwen3.5|397B-A17B|79.2|78.5|81.6|65.3|85.0|89.8|87.3|74.9|72.1|85.3|83.0|77.5|69.9|
> |GPT-5.2|-|71.0|85.0|85.1|29.0|74.6|91.2|84.8|81.0|72.1|26.0|74.1|73.1|75.5|
>
> Furthermore, we extended our LLaVA-GeP methodology to the modern Qwen3-VL architecture. Since its training data is not publicly available, we use the LLaVA-1.5 training data (mixed with our synthethic geometric data) to train the Qwen3-VL from scratch. As summarized in the table, the geometric perception enhanced model exhibits considerable gains across diverse downstream tasks. This proves that our proposed data synthesis engine is model-agnostic and that geometric perception remains a universal bottleneck across different generations of architectures.
>
> |Model|Average|General Visual Capability|Downstream Application|
> |-|-|-|-|
> |LLaVA-1.5-7B|49.2|61.5|36.8|
> |LLaVA-GeP|50.7|62.7|38.8|
> |Qwen3-VL-LLaVA-8B|63.5|72.9|54.1|
> |Qwen3-VL-GeP|64.8|74.2|55.4|

---

> > ### Author Rebuttal · Reviewer_Lzvq · 2026-04-04
> >
> > Thanks for the authors' detailed responses, which successfully addressed my concerns. I have no further questions. I will improve the score.

---

### Official Review · Reviewer_Wujm · 2026-03-12

**Soundness:** 3
**Presentation:** 3
**Significance:** 3
**Originality:** 2
**Overall Recommendation:** 4
**Confidence:** 3

**Summary:**

This paper introduces GePBench, a large-scale benchmark to evaluate the geometric perception of MLLMs. It includes 80K geometric figures and 285K multiple-choice questions across six aspects: location, size, existence, counting, reference, and relationship. Data is generated via a synthesis engine that creates structured textual descriptions of figures, rendered into images with Matplotlib. Questions are categorized into easy and hard based on shape count and visual noise. The paper tests 27 top MLLMs, finding significant deficiencies, especially in size and location tasks. The authors also propose LLaVA-GeP, trained on GePBench data, which improves performance on downstream tasks like medical imaging, chart understanding, and document analysis.

**Compliance With Llm Reviewing Policy:**

Affirmed.

**Final Justification:**

Thank the authors for the thorough rebuttal, which includes concrete experimental evidence.
The paraphrasing experiment convincingly addresses my template shortcut concern (Q2), the LLaVA-Chart control experiment effectively isolates the contribution of geometric perception data versus general synthetic augmentation (Q3), and the MAE analysis for counting tasks confirms that accuracy is a reliable proxy (Q4). I also appreciate the real-world evaluation on hand-drawn and scientific diagrams (Q1/W1), which demonstrates that synthetic-to-real transfer holds and that LLaVA-GeP generalizes well. My concerns are adequately addressed.

**Key Questions For Authors:**

1. Have you considered evaluating models on small set of real-world geometric images to validate that performance on GePBench's synthetic images correlates with real-world geometric perception?

2. The template-based question generation could introduce exploitable patterns. Have you tested whether models achieve higher accuracy through pattern matching to quantify the extent of shortcut learning?

3. For the LLaVA-GeP training, how much of the improvement on downstream tasks comes from general visual training data augmentation versus specifically geometric perception? Have you tried training with an equivalent amount of non-geometric synthetic visual data as a control?

4. For counting tasks, have you considered metrics that account for the magnitude of error?

**Limitations:**

Yes

**Strengths And Weaknesses:**

### Strengths

S1: Many benchmarks focus on high-level reasoning, but the basic ability to perceive geometric shapes, relationships, and attributes is often overlooked. Such foundational perception is crucial for higher-order tasks, and this is well-supported.

S2: The six evaluation aspects cover the key dimensions of geometric perception well, and the easy/hard split with controlled difficulty factors enables fine-grained analysis.

S3: The scale is impressive, which enables a statistically reliable evaluation.

S4: Downstream transfer experiments with LLaVA-GeP show that training on geometric perception data enhances performance on medical imaging, chart understanding, and scientific diagrams, emphasizing the importance of geometric perception.

S5: The error analysis provides actionable insights, identifying five concrete failure modes with quantitative distributions.

### Weaknesses

W1: The benchmark uses synthetic images rendered with Matplotlib, creating a domain gap from real-world geometric perception tasks. The shapes are clean, programmatically drawn figures that may not reflect the challenges of perceiving geometry in natural images, engineering drawings, or scientific figures. Although noise is added, it is artificial and doesn't mirror real-world degradation.

W2: The paper lacks analysis on whether models learn template shortcuts instead of genuine geometric perception. It also doesn't fully detail how distractors are generated to prevent trivial elimination.

W3: The claim that specialized geometric reasoning models don't outperform base models on GePBench suggests they neglect perception. Alternatively, GePBench's synthetic distribution may differ enough from training data to make the comparison unfair.

W4: The paper overlooks potential ceiling effects. If near-perfect human accuracy is possible and tasks are unambiguous by design, the benchmark may have limited longevity as models improve.

---

> ### Author Rebuttal · Authors · 2026-03-31
>
> Thank you for your appreciation of our well-supported motivation, fine-grained analyses, impressive evaluation scale and actionable insights. For the concerns, we detail our response below. We would appreciate it if you could let us know whether your concerns are addressed.
>
> $~$
>
> ---
>
> $~$
>
> > ***(Q1 & W1) Real-world geometric images.***
>
> **A:** Per Appendix E.2.2, we evaluated on real-world images (hand-drawn sketches, scientific diagrams from ScienceQA, AI2D, ChartXiv). Model rankings remain consistent across synthetic/real domains, with a moderate additional drop on real inputs. This confirms synthetic deficiencies persist and intensify in real-world inputs. Notably, LLaVA-GeP significantly outperforms its baseline (61.7% vs. 40.0%), demonstrating strong generalization of geometric perception learned from synthetic data.
>
> $~$
>
> ---
>
> $~$
>
> > ***(Q2 & W2) Template-based question generation.***
>
> **A:** To assess whether models exploit question templates as shortcuts, we conducted additional experiments in which an LLM paraphrased all benchmark questions while preserving semantic content. Performance comparison of representative models is provided in the following table.
>
> |Model|Original|Paraphrased|Difference|
> |-|-|-|-|
> |gpt-4o|63.2|64.5|1.3|
> |Qwen2.5-VL-7B|62.4|60.5|-1.9|
> |Qwen2.5-VL-72B|66.0|67.9|1.9|
> |InternVL3-8B|64.0|61.7|-2.3|
> |InternVL3-78B|71.0|68.9|-2.1|
>
> Despite expected minor fluctuations, model performance remains highly consistent across original and paraphrased questions. This stability indicates that MLLMs rely on genuine visual-linguistic understanding rather than template memorization.
>
> $~$
>
> ---
>
> $~$
>
> > ***(W2): Details on distractors to prevent trivial elimination.***
>
> **A:** As detailed in Appendix A.4, we implemented a rigorous distractor generation strategy to mitigate elimination-based shortcuts. For example, for counting and size-comparison tasks, distractors are intentionally calibrated to be numerically proximate to the correct answer yet discriminable (±1 for counts, ±0.15 for relative size ratios).
>
> $~$
>
> ---
>
> $~$
>
> > ***(Q3) Control experiment of non-geometric synthetic visual data.***
>
> **A:** To rigorously isolate the impact of our geometric perception data, we conducted a control experiment. We trained LLaVA-Chart using equivalent synthetic chart data from ChartVerse. We then evaluated both LLaVA-GeP and LLaVA-Chart across our suite of downstream tasks. The results indicate that while LLaVA-Chart predictably improves performance on chart and document understanding, it fails to generalize to other domains. In contrast, LLaVA-GeP achieves improvements across a diverse array of tasks, validating geometric perception data instill fundamental spatial awareness with broad transferability.
>
> |Model|Average|General VQA|Doc & Chart|Math|Medical|
> |-|-|-|-|-|-|
> |LLaVA-1.5-7B|49.2|61.5|38.5|19.8|43.2|
> |LLaVA-Chart|50.3|62.2|41.3|19.5|44.4|
> |LLaVA-GeP|50.7|62.7|40.7|21.1|45.3|
>
> $~$
>
> ---
>
> $~$
>
> > ***(Q4) Magnitude of error for counting tasks.***
>
> **A:** To quantify the magnitude of error in the counting tasks, we evaluated the Mean Absolute Error (MAE) and compared it against our standard Accuracy metric. We have also provided a visualization of the error distributions for representative models [here](https://ibb.co/G4Jz5Y5z).
>
> |Metric|GPT-4o|InterVL3-8B|InternVL3-78B|Qwen2.5-VL-7B|Qwen2.5-VL-72B|
> |-|-|-|-|-|-|
> |Accuracy|74.4|57.5|84.2|61.5|82.8|
> |MAE|0.325|0.526|0.167|0.494|0.197|
>
> The magnitude-aware analysis reveals a strong inverse correlation between Accuracy and MAE across all tested models, confirming that accuracy metric can be a reliable indicator of geometric counting capabilities.
>
> $~$
>
> ---
>
> $~$
>
> > ***(W3) Distributional difference between GePBench and geometric reasoning models.***
>
> **A:** We acknowledge the possibility of such distributional shifts. However, basic geometric shapes and their relative spatial relationships constitute the foundational visual elements underlying any complex geometric reasoning task. If models lose perceptual capability on these semantics-free, pure geometric figures, their strong performance in complex scenarios likely relies more heavily on textual logical shortcuts rather than genuine visual understanding. We will add an objective discussion in the Discussion section.
>
> $~$
>
> ---
>
> $~$
>
> > ***(W4): Ceiling effects.***
>
> **A:** Although humans achieve 99.3% accuracy, even the powerful MLLMs (e.g., GPT-4o) still exhibit a substantial 30% performance gap compared to humans. In the journey toward general-purpose multimodal large models, the phenomenon where tasks are extremely simple for humans yet extremely difficult for AI precisely exposes critical blind spots in the alignment capabilities of current models. Until MLLMs bridge this gap, GePBench will serve as an essential diagnostic dataset, driving the community to optimize fundamental visual perception.

---

> > ### Author Rebuttal · Reviewer_Wujm · 2026-04-04
> >
> > Thank you for the thorough rebuttal with concrete experimental evidence.
> >
> > The paraphrasing experiment convincingly addresses my template shortcut concern (Q2), the LLaVA-Chart control experiment effectively isolates the contribution of geometric perception data versus general synthetic augmentation (Q3), and the MAE analysis for counting tasks confirms that accuracy is a reliable proxy (Q4). I also appreciate the real-world evaluation on hand-drawn and scientific diagrams (Q1/W1), which demonstrates that synthetic-to-real transfer holds and that LLaVA-GeP generalizes well. My concerns are adequately addressed.

---

### Decision · Program_Chairs · 2026-04-30

**Decision:**

Accept (regular)

**Comment:**

The paper studies the MLLM's ability for recognizing geometric shapes and their spatial relationships. To study this, this paper proposed GePBench, a novel benchmark specifically designed to assess the geometric perception capabilities of MLLMs. With extensive evaluations, even the current state-of-the-art MLLMs exhibit significant deficiencies in geometric perception tasks. This is a good finding and a good benchmark.

All the reviewers recommended weak acceptance and agreed that the rebuttal fully addressed their concerns. I agree with them and vote for acceptance.